# The Promotion of Mechanical Properties by Bone Ingrowth in Additive-Manufactured Titanium Scaffolds

**DOI:** 10.3390/jfb13030127

**Published:** 2022-08-26

**Authors:** Changning Sun, Enchun Dong, Jiayu Chen, Jibao Zheng, Jianfeng Kang, Zhongmin Jin, Chaozong Liu, Ling Wang, Dichen Li

**Affiliations:** 1State Key Laboratory for Manufacturing System Engineering, School of Mechanical Engineering, Xi’an Jiaotong University, Xi’an 710054, China; 2National Medical Products Administration (NMPA) Key Laboratory for Research and Evaluation of Additive Manufacturing Medical Devices, Xi’an Jiaotong University, Xi’an 710054, China; 3Institute of Orthopaedic & Musculoskeletal, University College London, Royal National Orthopaedic Hospital, London HA7 4LP, UK; 4Jihua Laboratory, Foshan 528200, China; 5School of Mechanical Engineering, University of Leeds, Leeds LS2 9JT, UK

**Keywords:** additive manufacturing, porous scaffolds, osseointegration, finite element analysis

## Abstract

Although the initial mechanical properties of additive-manufactured (AM) metal scaffolds have been thoroughly studied and have become a cornerstone in the design of porous orthopaedic implants, the potential promotion of the mechanical properties of the scaffolds by bone ingrowth has barely been studied. In this study, the promotion of bone ingrowth on the mechanical properties of AM titanium alloy scaffolds was investigated through in vivo experiments and numerical simulation. On one hand, the osseointegration characteristics of scaffolds with architectures of body-centred cubic (BCC) and diamond were compared through animal experiments in which the mechanical properties of both scaffolds were not enhanced by the four-week implantation. On the other hand, the influences of the type and morphology of bone tissue in the BCC scaffolds on its mechanical properties were investigated by the finite element model of osseointegrated scaffolds, which was calibrated by the results of biomechanical testing. Significant promotion of the mechanical properties of AM metal scaffolds was only found when cortical bone filled the pores in the scaffolds. This paper provides a numerical prediction method to investigate the effect of bone ingrowth on the mechanical properties of AM porous implants, which might be valuable for the design of porous implants.

## 1. Introduction

Additive-manufactured (AM) orthopaedic implants with porous structures have received extensive attention in recent years because they not only weaken the stress shielding by reducing the equivalent elastic modulus of the implants [1,2] but promote long-term osseointegration through bone ingrowth in the porous structure [3,4]; thus, it appears as a promising solution to solve the long-term aseptic loosening of the metal orthopaedic implant. Metal AM technologies, mainly achieved by powder bed fusion (PBF), have become the main technique for manufacturing porous implants [5,6,7] and have been used in clinical research, such as in dental implants [8], artificial vertebral bodies [9], pelvic reconstruction [10], and acetabular cups [11], some of which have been commercialized.

The porous structures of orthopaedic implants are mainly built by periodic arrays of representative volume elements (RVEs) [12]. As the basic units of the porous implants, the topology and dimensions of the RVEs are essential for both mechanical properties and bone ingrowth. In the research of AM porous implants, the mechanical properties and osseointegration of a variety of different RVEs, such as diamond, orthotropic cubic, honeycomb, and triply-periodic minimal surface, have been investigated with diamond structures receiving the most interest in research [13,14,15] and clinical applications [9,10]. Porosity and pore size are commonly regarded as key parameters, and their influences on osseointegration was conclusive; it was generally accepted that porosity and pore size in the ranges of 50~85% and 500–800 μm, respectively, facilitate bone ingrowth into porous structures [16]. However, most studies have mainly focused on the promotion of bone ingrowth in porous structures with investigations on the mechanical-bonding strength of metal AM porous structures with bone tissue being rare. The available studies characterizing the tensile-bonding strength and the shear-bonding strength of the bone–implant interface are limited. The tensile-bonding strength between titanium alloy (Ti alloy) scaffolds with RVEs of diamond architecture with different pore sizes was measured by Taniguchi et al. [15] who found that a pore size of 600 μm was optimal because it demonstrated the highest bonding strength compared with 300 μm and 900 μm. Hara et al. [17] investigated the shear bonding between porous Ti alloy and bone by pushing the cylindrical scaffolds out along the axis. A similar method was used to measure the shear bonding between AM polyether–ether–ketone (PEEK) scaffolds and bone tissue by Zheng et al. [18]. Koolen et al. [19] studied the torsional-bonding forces at the interface in the reconstruction of a tibial segment by porous Ti alloys where the torsion at the interface was essentially one of the manifestations of shear. Compared with the tension, shear bonding was more in line with the actual clinical situation, such as femoral stem, screws, and bone fracture plates, yet very few orthopaedic implants are supposed to be subject to tensile loads on their interface with the surrounding bone.

As new bone tissue grows into porous implants, it fills the voids in the porous structure, which enhances the mechanical properties of the osseointegrated porous structure by changing the porous structure to a composite material consisting of hard and soft materials. There are very limited direct data from animal experiments available in the literature as to how bone ingrowth affects the mechanical properties of a porous structure. Ren et al. [20] observed increases in the elastic modulus of porous hydroxyapatite (HA) in animal experiments from 38 MPa to 380 MPa at 12 weeks after implantation and developed a three-dimensional (3D) finite element (FE) model to simulate the promotion of regenerated bone to the porous HA. In animal experiments of HA/PEEK composite AM scaffolds, Zheng et al. also found that the strength of composite scaffolds with an HA content of 40 wt.% was enhanced from 16 MPa to 24 MPa by the bone ingrowth. Arahira et al. [21] also identified an enhancement in the elastic modulus and the strength of porous bioceramic scaffolds by the proliferation of osteoblasts in cell experiments. However, due to the limited number and specificity of test samples, animal experiments were inadequate in quantifying the mechanobiology of the scaffold in vivo. Thus, in vitro experiments became an alternative to quantify the effect of bone ingrowth on the mechanical performance of the porous implants. Zadpoor et al. [22] and Rodríguez-Montaño et al. [23] simulated the mechanical properties of AM metal scaffolds filled with resin through both the FE method and experiment; their results illustrated that the fatigue performance of the metal scaffolds was improved by two to seven fold, even filled by resin with an elastic modulus lower than that of cortical bone.

Although there have been some studies on the mechanical properties of osseointegrated porous structures, the contradiction between the realism of in vitro simulations and the quantifiability of animal experiments has limited further developments in the research. The contradiction was that it is difficult to control the pattern, volume, and type of bone tissue in the porous structures in any animal experiments and to establish a quantitative understanding of the effect of bone ingrowth on the mechanical properties of porous structures. In in vitro simulations, for example, when filling the porous structure with resin, it was difficult to reflect the real distribution patterns of bone tissue in the void of the porous structures. In fact, it is very difficult for bone tissue to completely fill the void of a small RVE in porous implants. However, the in vitro simulations were mostly based on complete fill patterns due to the resin-filling method. One of the crucial points in the design of the porous implant is the balance between reducing the stress shielding via increasing porosity and decreasing the porosity to safeguard the long-term mechanical properties of the porous structure. Thus, a more accurate prediction of the effect of bone ingrowth on the mechanical properties of porous structures by developing accurate FE models of AM scaffolds and osseointegration scaffolds would support the development of the design methodology of porous orthopaedic implants based on the long-term properties of osseointegrated implants. It is possible to conceptualize the morphological and properties of bone tissue in the scaffolds in FE models; therefore, more quantitative results could be obtained than from the existing animal experiments and in vitro simulations.

In this study, the effect of bone ingrowth on the mechanical properties of the osseointegrated scaffolds was investigated through animal experiments and computational simulations. In the in vivo animal experiments, scaffolds with RVEs of body-centred cubic (BCC) were employed, and the osseointegration in BCC scaffolds was compared with that in scaffolds with RVEs of diamond structure. An FE model of the osseointegrated BCC scaffold was developed and calibrated using the results of in vivo experiments. The effect of bone ingrowth within the scaffolds on the mechanical properties was systematically investigated by the FE model.

## 2. Materials and Methods

### 2.1. Fabrication of Scaffolds

Cylindrical scaffolds with RVEs of BCC and diamond porous architecture (Figure 1A) were fabricated using laser powder bed fusion (LPBF) technology (BLT-S200, BLT, Xi’an, China), which is also known as selective laser melting (SLM), using Ti6Al4V powder. The scaffolds are presented in Figure 1B. The design and manufacturing parameters of the scaffolds are shown in Table 1 and Appendix A Table A1, respectively. The designed relative densities, *φ_v_*, of both scaffolds were made the same through the use of different strut sizes in different scaffolds.

### 2.2. In Vivo Animal Experiments

#### 2.2.1. Surgical Procedure

The protocol of the animal experiment was approved by the Ethics Committee of Xi’an Jiaotong University Health Science Center (approval number: 2020–1136). The principles of the 3Rs (replacement, reduction, and refinement) were followed when designing the animal experiments. The scaffolds with different porous structures were sterilized in a moist heat cabinet and then implanted into the lateral femoral condyles (Figure 1C–E) of 12 male adult New Zealand white rabbits with an average weight of 2.94 ± 0.36 kg, which were purchased from Xi’an Jiaotong University Health Science Center; a total of 34 scaffolds were implanted with one scaffold on each leg only. During the surgeries, the rabbits were anaesthetized with the intravenous injection of pentobarbital sodium (30 mg/kg) in saline. Cefazolin sodium (50 mg/kg) was injected intramuscularly daily for six days after the surgery to avoid infection. The experimental animals were euthanized at two and four weeks post-operatively, and the numbers of samples retrieved at different weeks are presented in Appendix A Table A2. Twenty-four scaffolds were retrieved in total.

#### 2.2.2. Radiographic Assay

Micro-CT (Y. Cheetah, YXLON, Germany) was used to quantify the bone volume fraction of the scaffolds retrieved two and four weeks post-operatively (*n* = 4 for each group with different RVEs and different weeks) on the day the experimental animal was sacrificed, according to Equation (1).
(1)$$P=\frac{V_{bone}}{V_{void}}$$
where *p* is the bone volume fraction, *V_bone_* is the volume of the bone tissue in the void of the scaffolds, and *V_void_* is the volume of the void in the scaffolds.

The retrieved scaffolds were scanned by micro-CT with a source voltage of 80 kV and beam current of 500 μA. The scanned images were obtained from VG Studio (Version 2.1, Or3D, Wrexham, UK), and beam-hardening correction was used to decrease metal artifacts. The threshold of the bone tissue was set to 300~1300 Hounsfield Unit (HU), and the threshold of scaffolds was higher than 1300 HU. The volume of the bone tissue and void of scaffolds were finally calculated based on the thresholds.

#### 2.2.3. Histological Characterization

Formalin-fixed tissues of the scaffolds retrieved two and four weeks post-operatively were paraffin-embedded and sectioned using a hard tissue microtome (SP 1600, Leica, Wetzlar, Germany) and polished to a thickness of 40~60 μm (*n* = 2 for each group with different RVEs at two and four weeks). All the slices were stained using Van Gieson (VG) and observed with an inverted fluorescence microscope (DM 4000B, Leica, Wetzlar, Germany).

#### 2.2.4. Biomechanical Test

The shear-bonding strength between the cylindrical scaffolds and the host bone (*n* = 4 for two and four weeks post-operatively) was measured according to the principle shown in Figure 1F. The retrieved sample was fixed in the jig with self-curing denture acrylic (Unifast, Tokyo, Japan), as presented in Figure 1G, and care was taken so that the axes of the cylindrical scaffolds were perpendicular to the bottom surface of the jig. The scaffold was pushed out along the axis by a metal pressure head at a speed of 1 mm/min with a universal mechanical-testing machine (MTS 880, MTS, Eden Prairie, MN, USA). The maximum force during the push-out was defined as the maximum shear load between the bone and scaffolds. The compression properties of the pushed-out scaffolds were tested immediately to investigate the effect of the newly generated bone on the mechanical properties of the scaffolds. For comparison, the compressive mechanical properties of the blank scaffolds (*n* = 6 for scaffolds with RVEs of BCC and diamond) with different RVEs were also tested with the universal mechanical-testing machine (MTS 880, MTS, Eden Prairie, MN, USA) at a speed of 1 mm/min.

#### 2.2.5. Statistical Analysis

All the experimental data are expressed as the means ± standard deviation (SD). Statistical analysis was performed with one-way ANOVA using SPSS software (Version 17.0, IBM, Armonk, NY, USA) and Tukey’s test. Statistically significant differences were defined as *p* < 0.05 (*) and *p* < 0.01 (**).

### 2.3. Finite Element Analysis

#### 2.3.1. Finite Element Analysis of the Blank Scaffold

Predicting the mechanical properties of porous scaffolds by the designed model was proven to be inaccurate with the 3D model directly reconstructed from CT images being very difficult to use in FE analysis due to the numerous small features or defects in the reconstructed model. A compromise was established by fitting the cross-sections of the structures of the 3D model BCC scaffolds reconstructed from micro-CT images (Figure 2A) and then creating a 3D model based on the fitted cross-section. The theoretical cross-sectional shape of the strut of the BCC scaffold is circular; thus, the real cross-section of the strut was fitted by the inscribed circle and least-square circle, and the 3D model of the BCC scaffold was reconstructed based on the fitted cross-section (Figure 2B).

The equivalent mechanical properties of the BCC scaffolds fitted by the above methods and reconstructed directly from CT images were predicted by FE analysis in Abaqus (Version 6.14, Dassault, Vélizy-Villacoublay, France) to evaluate the effectiveness of the different fitting methods. The calculation method of the mechanical properties, including equivalent elastic modulus and equivalent yield strength, of the porous scaffolds, as well as the boundary conditions and the loads used in the FE analysis, have been described in detail in previous studies [24]. Briefly, the bottom surface of the scaffold model was fixed, and the compressive load, *F*, was applied on the top surface of the scaffold. Elements of the 10-node quadratic tetrahedron (C3D10) were used in the FE analysis with a global mesh size of 0.1 mm and local mesh size of 0.05 mm at the connection region of the structure of the BCC scaffold according to the mesh sensitivity check. The equivalent yield stress was determined by the yield point of the equivalent stress–strain curve, and the equivalent elastic modulus (*E_eff_*) was calculated using Equation (2).
(2)$$E_{eff}=\frac{F\cdot \Delta h}{h\cdot A_{ncs}}$$
where *F* is the load that is applied on the top surface of the scaffold, Δ*h* is the displacement of the top surface of the scaffold, *h* is the height of the cylinder scaffold, and *A_ncs_* is the nominal cross-section area.

The FE model struggles to reflect the effect of the surface and internal defects in the PBF parts on the mechanical properties for which the predicted elastic modulus of the scaffolds was usually much higher than the experimental results; thus, previous studies have employed correction methods based on the Gibson–Ashby model [24]. More accurately predicted results were obtained by multiplying the elastic modulus of the material in the FE model by a correction factor, *C*, reflecting the effect of internal defects on the mechanical properties.

#### 2.3.2. Finite Element Analysis of the Osseointegrated Scaffold

The mechanical properties of the osseointegrated scaffold were simulated by FE analysis according to the pattern of bone ingrowth in the void of the scaffold observed by micro-CT. The regenerated bone tissue adhered to the surface of the struts in the scaffolds according to micro-CT images and histological staining; thus, an FE model in which cancellous bone completely wrapped the scaffolds was developed based on this phenomenon, as shown in Figure 2C, aiming to evaluate the osseointegration scaffold FE model. The volume of the bone tissue in the FE model was determined by that in the BCC scaffolds retrieved four weeks after implantation. The equivalent compressive elastic modulus and equivalent yield strength of the osseointegrated scaffold were calculated by FE analysis. Comparisons between the predicted mechanical properties of the osseointegrated scaffold model and the measured values were employed to evaluate the FE model.

#### 2.3.3. Prediction of Bone Ingrowth on the Mechanical Properties

The periphery of the cylinder scaffolds was expected to be filled by the regenerated bone tissue over time. Here, we hypothesize that the bone tissue in the scaffold first wraps the surface of the micro-structs and then gradually fills the pores of the scaffold from the outside to the inside. For a deeper understanding of the effect of bone ingrowth on the mechanical properties of the osseointegrated scaffold, as shown in Figure 2D, a conceptual model was developed to investigate the effect of morphology and type of bone tissue in the scaffold on the mechanical properties. In the conceptual model, the width ($t=\frac{1}{3}R$ or $t=\frac{2}{3}R$) of the filled region of the scaffold and the volume ($V=\frac{1}{3}V_{c}$ or $V=\frac{2}{3}V_{c}$) of the wrap region of the scaffold were defined as the parameters related to the morphology, where *R* is the radius of the cylinder scaffolds, and *V_c_* is the volume of the void in the scaffolds except for the filled region. It was assumed that the pores within the periphery region of the cylinder scaffolds were completely filled with bone tissue, while the strut in the central region was uniformly wrapped. Additionally, the type of bone tissue, i.e., cortical or cancellous, was set as another variable. To fully investigate the influence of *t*, *V*, and the type of the bone tissue within the scaffolds on the mechanical properties of the osseointegrated scaffolds, the L_4_(2^3^) orthogonal test with three factors and two levels for each factor was designed, and in total, four simulations were run. The details of the orthogonal test are presented in Appendix A, Table A3. FE models of a quarter cylindrical scaffold were used for the analysis (Figure 2E,F) to reduce the computational time. The material properties involved in the FE analysis of this study are summarized in Appendix A Table A4 and Appendix A Figure A1.

## 3. Results

The VG-stained histological sections of the scaffolds retrieved at two and four weeks are presented in Figure 3. The bone ingrowth properties in the scaffolds with different RVEs were similar with the regenerated bone tissue only observed at the outline of the scaffolds at two weeks. At four weeks after implantation, the bone tissue was found in the pores of the scaffolds. Bone tissue was found crawling over the surface of the micro-structs of the scaffolds, growing deeper into the pores in the central area of the scaffolds.

Representative 2D and 3D micro-CT images of the scaffolds with different RVEs at two and four weeks after implantation are shown in Figure 4. The bone volume fractions in the BCC scaffolds were significantly higher than those of the diamond scaffolds, although no significant differences were found four weeks after implantation.

Typical force-displacement curves during the push-out test of the retrieved scaffolds are presented in Figure 5A; a similar trend was shown for the scaffolds with different RVEs. The maximum push-out force is summarized in Figure 5B, demonstrating that the implantation time had a significant effect on the push-out force, whereas the architecture hardly affected it. The equivalent elastic modulus and equivalent compressive strength of the pushed-out scaffolds are presented in Figure 5C,D, respectively. For the BCC scaffolds, the bone ingrowth did not affect the mechanical properties, although slight decreases in both the elastic modulus and compressive strength of diamond scaffolds were observed.

The diameters of the micro-structs of the CT-reconstructed BCC scaffolds fitted by the inscribed circle and least-square circle were 132 μm and 145 μm, respectively. The predicted equivalent elastic modulus and compressive strength of the inscribed circle fitted model, least-square-circle model, and CT-reconstructed model are summarized in Figure 6A,B in comparison with the experimental results. The relative error of elastic modulus simulated by the inscribed circle fitted model was 153%, which was much lower than that of the least-square circle, whereas the CT model had a similar relative error of 189%. A correction coefficient, *C*, was calculated using Equation (3), and the corrected elastic modulus of PBF-Ti6Al4V was calculated using Equation (4).
(3)$$C=\frac{E_{in}}{E_{ex}}$$
(4)$$E_{Ti}^{\prime }=E_{Ti}\times C$$
where *E_in_* is the elastic modulus calculated by the inscribed circle fitted model, *E_in_* = 7.54 Gpa, *E_ex_* is the elastic modulus measured from the experiment, *E_ex_* = 1.98 Gpa, and *C* is the correction coefficient, *C* = 0.39. *E_Ti_* is the elastic modulus of Ti6Al4V, *E_Ti_* = 110 Gpa, and $E_{Ti}^{\prime }$ is the correct elastic modulus of PBF-Ti6Al4V, $E_{Ti}^{\prime }$ = 42.9 GPa.

The mechanical properties predicted by the corrected elastic modulus of PBF-Ti6Al4V are presented in Figure 6C,D. The relative errors of elastic modulus and compressive strength simulated by the inscribed circle fitted model decreased to 3.3% and 5.7%, respectively.

The distribution of von Mises stress and displacement of the integrated scaffolds with $t=\frac{1}{3}R$ and $V=\frac{1}{3}V_{c}$ are presented in Figure 7A,C–F as demonstrations, respectively. The predicted elastic modulus and yield compressive strength of the osseointegrated scaffolds with different values of *t* and *V* are shown in Figure 7G,H. The type of bone tissue was indicated to have the greatest effect on the mechanical properties of the integrated scaffolds, whereas the volume of the wrap region, *V*, had hardly any effect. The mechanical performance of the scaffolds with a bone volume fraction of 15.5%, which was the actual bone volume fraction of BCC scaffolds retrieved at four weeks, was also calculated. The predicted elastic modulus and yield strength were 3.23 GPa and 91.50 MPa, respectively.

## 4. Discussion

The effect of bone ingrowth on the mechanical properties of the porous structures has not been adequately quantified. In this study, the osseointegration of the scaffolds with RVEs of BCC and diamond structure was compared through animal experiments. An FE model for the osseointegrated AM porous metal scaffold was developed and was calibrated by comparing it with the results of the in vivo experiments. A conceptual model was proposed to describe the morphology and to predict the mechanical properties of bone tissue regenerated in the AM scaffolds by which the effects of bone ingrowth on the mechanical properties of osseointegration were systematically investigated using the FE model. A feasible numerical simulation method to study the mechanical properties of osseointegrated scaffolds, which was proven to solve the contradiction between the specificity of animal experiments and the realism of in vitro experiments, was developed in this study. Through the simulation, we found that bone ingrowth had a significant enhancement on the AM Ti alloy scaffold only when the cortical bone filled the pores of the scaffolds.

In animal experiments, the bone volume fraction in the BCC scaffolds was significantly higher than that of diamond scaffolds commonly used in clinical practice at the early stage (two weeks) after implantation. It was presumably because the higher specific surface area of BCC scaffolds at the same porosity compared with diamond scaffolds, which is more favourable for cell crawling over the surface of the micro-structs of the scaffolds at the early stage. No significant differences were observed in the shear-bonding strength at the bone–scaffold interface. The shear-bonding strength was composed of two parts, namely the micro-lock effect [17,25] caused by bone ingrowth of the scaffold pores and the friction between the scaffold and the surrounding bone. At the early stage of the push-out experiment, the push-out force was affected by the synergy of the micro-lock and the friction, and the maximum push-out force was reached when the micro-lock effect was strongest. Subsequently, the biological fixation of the bone–scaffold interface was damaged, and the push-out force was mainly contributed by the friction. With the increasing displacement, the contact area of the bone and the scaffold decreased, and the push-out force decreased accordingly. Decreases in the mechanical properties, including both equivalent elastic modulus and yield strength of diamond scaffolds, were observed in the compressive test of the pushed-out scaffolds. Despite the lack of direct evidence, the removal of the loosely connected metal particles from the surface of the scaffolds [26] in vivo was presumed to be one of the potential reasons for the decreased mechanical properties.

In FE analysis, the predicted mechanical properties of the model fitted by the inscribed circle were close to those of the CT-based model, while the CT-based model was time-consuming and computationally expensive during modelling. The predicted equivalent elastic modulus of the scaffolds using the theoretical elastic modulus of the Ti alloy was much larger than that of the experimental value, which is consistent with contemporary studies [27,28]. The surfaces of scaffolds fabricated by PBF technology are usually rough and exhibit many internal defects [29]. The inscribed circle was used to fit the cross-section of the micro-structs of the scaffold to eliminate the influence of the rough surface morphology on the mechanical properties of the scaffold. However, it is difficult to directly quantify the random defects inside the micro-structs. Therefore, according to Wang et al.’s study [24], high accuracy was achieved in predicting the equivalent mechanical properties of the scaffolds by modifying the elastic modulus of the PBF Ti alloy.

In this study, three parameters were employed in the FE prediction of mechanical properties of osseointegrated BCC scaffolds, namely the volume, the thickness, and the type (cortical or cancellous bone) of regenerated bone tissue surrounding the micro-structs of the scaffold. It was observed that the mechanical properties of the scaffold were significantly improved with the cortical bone ingrowth. For the two conceptual distribution patterns of bone tissue, pores filled by bone tissue exhibited an improvement in mechanical properties of scaffolds, whereas bone tissue only wrapped on the surface of the micro-structs hardly contributed to the mechanics of scaffolds. In animal experiments, however, the micro-CT and compression experiments were only conducted for scaffolds implanted for two and four weeks. The short implantation time led to bone tissue being hard to fill in the whole scaffold or even only the outer-layer pores. Therefore, both the compression test and the FE simulation results of the osseointegrated scaffolds showed no significant difference in mechanical properties compared with the blank scaffolds.

It was found that the mechanical properties of the osseointegrated scaffolds were not significantly improved in either the experiment or FE simulation because the elastic modulus of the PBF Ti alloy was much larger than that of bone tissue, especially for the cancellous bone filling the scaffold. Zheng et al. [18] found that the yield strength of the hydroxyapatite (HA)/polyether–ether–ketone (PEEK) composite scaffold was significantly higher than that of the blank scaffold at 12 weeks after implantation, while no significant enhancement was observed at four weeks after implantation. It may be related to some of the pores completely filled by bone tissue 12 weeks after implantation. It illustrated that the bone tissue could only play a bearing role when it fills a pore.

Understanding the effect of bone ingrowth on the mechanical properties of the osseointegrated scaffolds is a basis for the design of porous orthopaedic implants. In the future, the FE model developed in this study could be employed to investigate the mechanical properties of porous implants affected by bone tissue regeneration in long-term animal experiments. Better mechanical matching between porous implants and the host bone would be obtained by designing porous implants based on the expected mechanical properties of the osseointegration porous structure.

In this study, the assumptions of bone ingrowth in the scaffolds, such as wrapping and filling patterns, were used in the FE model. However, the morphology of natural bone tissue in scaffolds is irregular, and it is difficult for bone tissue to uniformly wrap the surface of micro-struts or to completely fill the pores of the scaffolds. Therefore, limitations in the assumptions made in this paper regarding the patterns of bone ingrowth. Some advanced experimental techniques [30], such as nanoindentation, atomic force microscopy, and quantitative ultrasound, can be used to measure the real mechanical properties of bone tissue near the bone–implant interface; thus, future research should focus on obtaining accurate properties of bone tissue in the osseointegration. In terms of computational simulation, multi-scale, time-varying bone tissue models [31] should be developed in the future to obtain more accurate predictions of the mechanical performance of osseointegration scaffolds. Another limitation is that the reason for the decreased mechanical properties of the diamond scaffolds in vivo was not fully investigated in this study, which should be investigated in the future.

## 5. Conclusions

The promotion of bone regeneration on the mechanical properties of AM Ti alloy scaffolds was investigated through in vivo experiments and numerical simulations. The influence of the type and morphology of bone tissue in the BCC scaffolds on the equivalent elastic modulus and equivalent compressive yield strength was investigated by FE models of osseointegrated scaffolds which were calibrated by comparison with the mechanical testing results. We found that for a porous structure composed of Ti alloys due to their extremely high stiffness and strength, significant promotion in its mechanical properties only occurred in the case of cortical bone filling the pores of the metal scaffolds. This study provides a numerical prediction method for investigating the effect of bone ingrowth on the mechanical properties of porous implants and is a potential basis for the future design of porous implants.

## Figures and Tables

**Figure 1 jfb-13-00127-f001:**
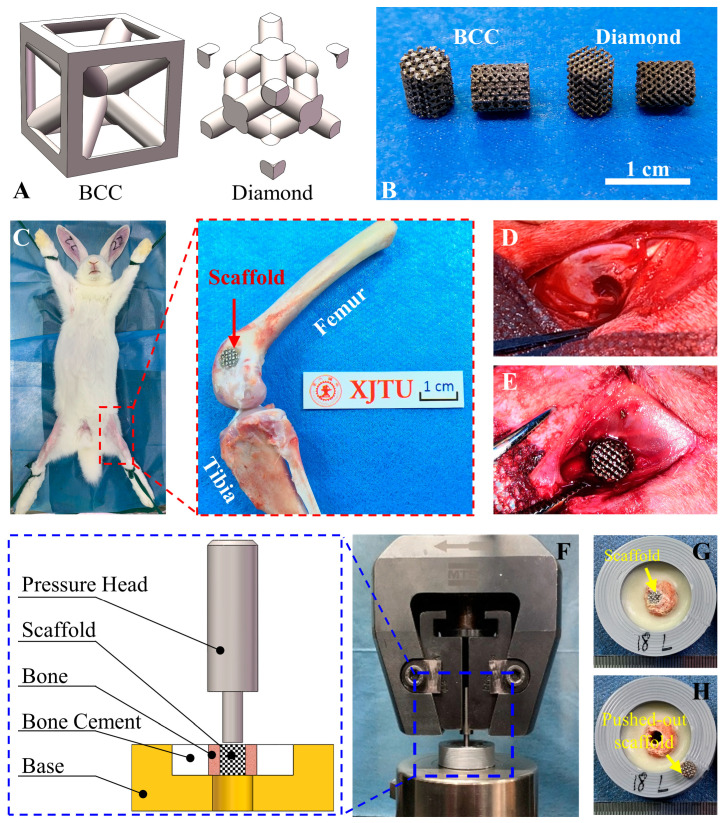
Fabrication and animal experiment of AM scaffolds. (**A**) RVEs of BCC and diamond porous architecture. (**B**) Photographs of the scaffolds. (**C**) The location of the implanted scaffolds. Intraoperative photographs of (**D**) bone defect and (**E**) implanted scaffolds. (**F**) Push-out test of the scaffolds. (**G**) The scaffolds in the surrounding bone. (**H**) Pushed-out scaffolds.

**Figure 2 jfb-13-00127-f002:**
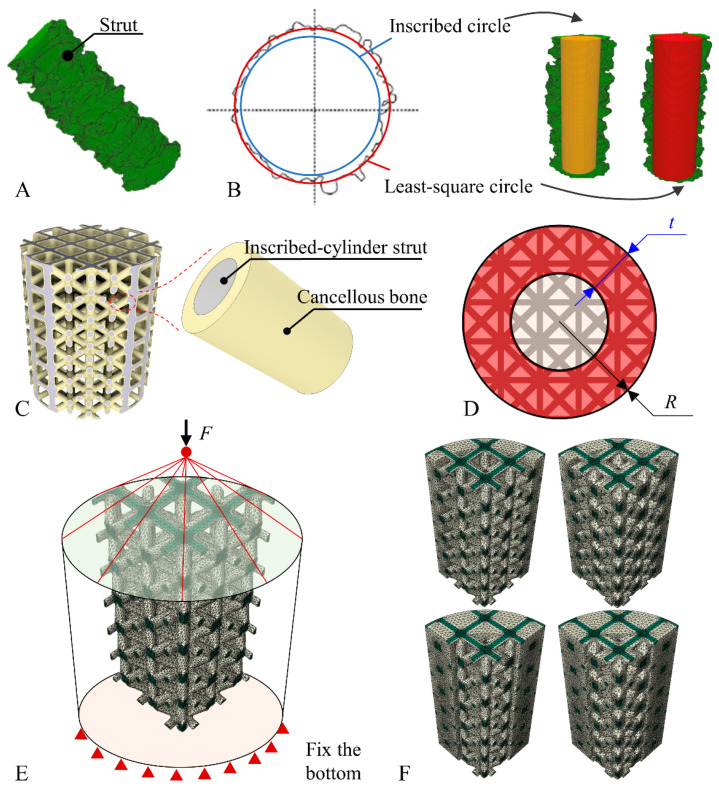
Reconstruction of the 3D model of BCC scaffolds and FE analysis. (**A**) CT-scanned micro-strut of the BCC scaffolds. (**B**) The cross-section of the strut was fitted using an inscribed circle and least-square circle. (**C**) Three-dimensional model of the fitted scaffold with surface wrapped in cancellous bone. (**D**) Schema of the BCC scaffold with the periphery (red area) filled with bone tissue and the surface of struts in the central area wrapped in bone tissue. (**E**) Load and boundary conditions of the FE model of a quarter cylindrical osseointegrated scaffold. (**F**) FE models simulating different bone ingrowth patterns in the scaffolds.

**Figure 3 jfb-13-00127-f003:**
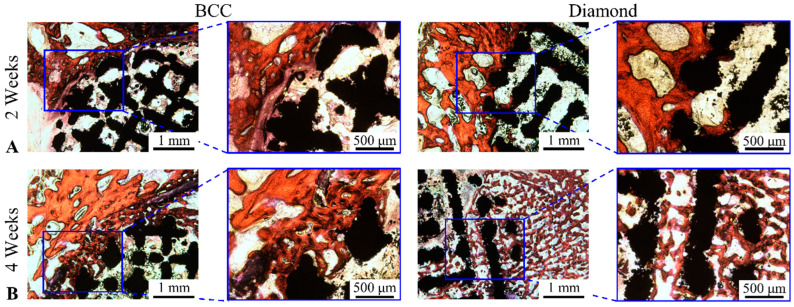
Van Gieson staining of histological sections of scaffolds with different RVEs. (**A**) BCC; (**B**) diamond (white: void; black: scaffolds; red: bone tissue).

**Figure 4 jfb-13-00127-f004:**
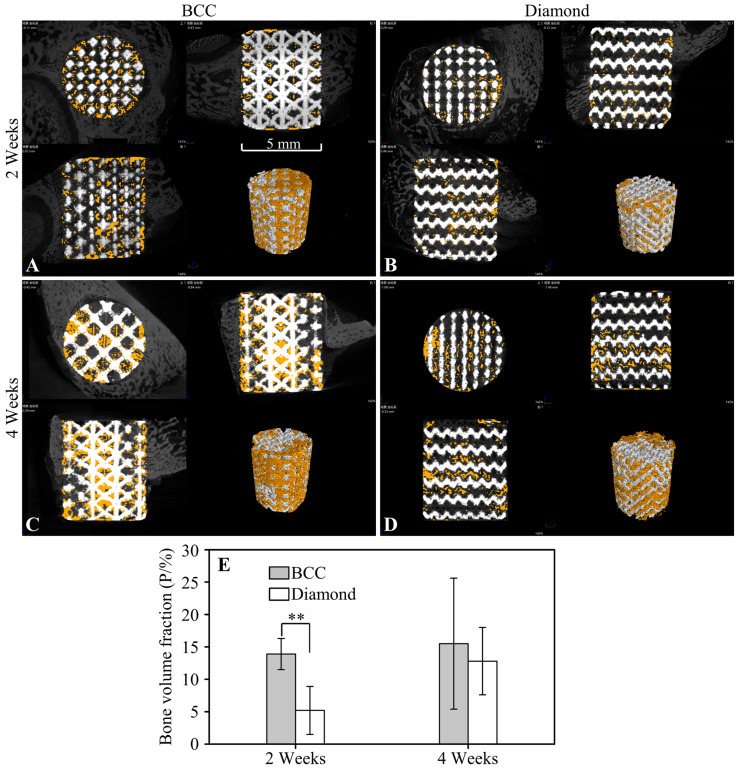
Results of the radiographic assay. (**A**–**D**) Micro-CT images and reconstructed 3D model of the retrieved scaffolds with RVEs of BCC and diamond at two and four weeks post-operatively (white: scaffolds; orange: bone tissue). (**E**) A summary of the bone volume fraction of different groups. (** *p* < 0.01).

**Figure 5 jfb-13-00127-f005:**
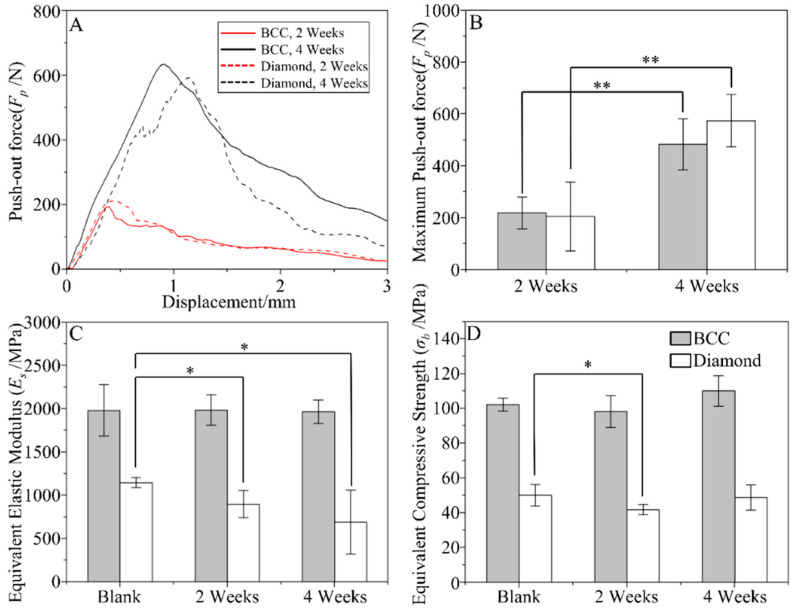
Results of the biomechanical test. (**A**) Typical force-displacement curves during pushing-out and (**B**) maximum push-out force of different RVEs at two and four weeks, (**C**) equivalent elastic modulus, and (**D**) equivalent compressive strength of the scaffolds. (* *p* < 0.05, ** *p* < 0.01).

**Figure 6 jfb-13-00127-f006:**
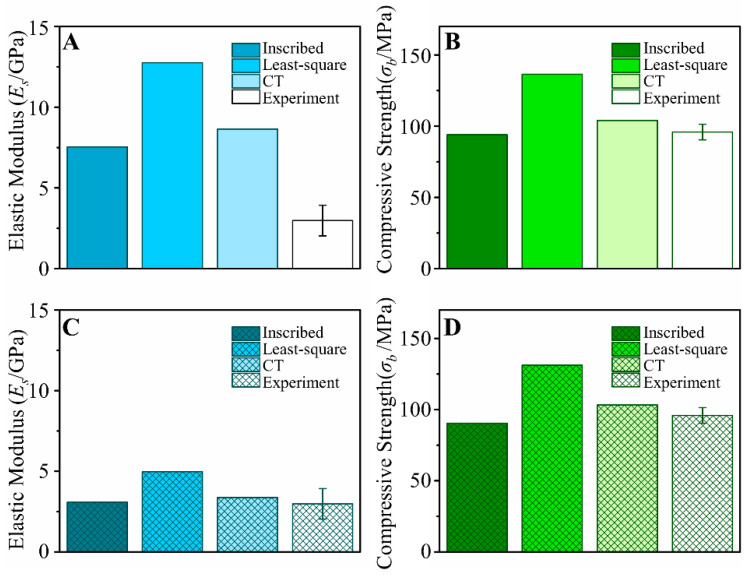
Predicted mechanical properties constructed by different methods. (**A**) Uncorrected elastic modulus, (**B**) uncorrected compressive strength, (**C**) corrected elastic modulus, (**D**) corrected compressive strength.

**Figure 7 jfb-13-00127-f007:**
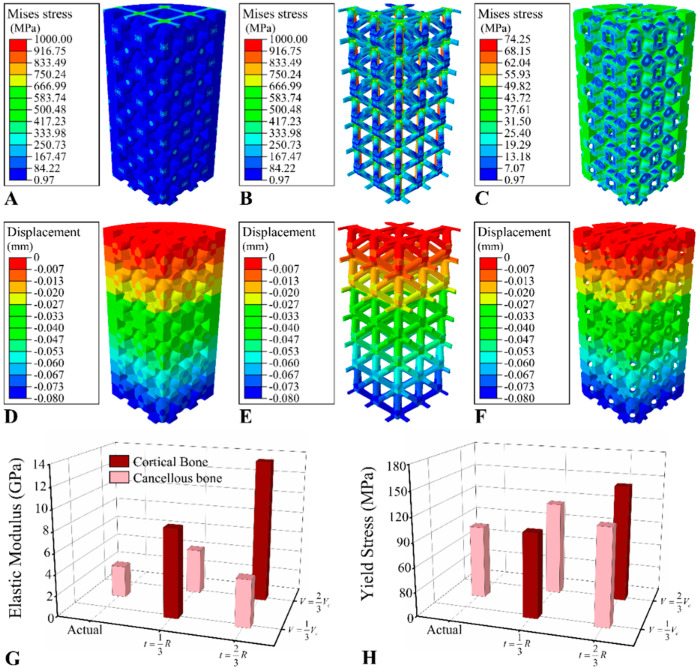
The FE results of the BCC scaffolds with bone tissue. The distribution of von Mises stress of (**A**) scaffold and bone tissue, (**B**) scaffold, and (**C**) bone tissue. The distribution of displacement along the axis of the scaffold of (**D**) scaffold and bone tissue, (**E**) scaffold, (**F**) bone tissue. (**G**) The elastic modulus and (**H**) compressive yield stress predicted by the FE model of integrated scaffolds with different filled regions and wrap regions.

**Table 1 jfb-13-00127-t001:** The design parameters of the scaffolds.

RVE	Geometry of Scaffolds	Strut Diameter (*d*/mm)	Size of RVEs (*a*/mm)	Surface Area/mm^2^	Relative Density (*φ_v_*/%)
BCC	*φ*5 *× h*6 mm	0.2	1	750.2	25
Diamond		0.25		452.0	

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
