# Peer review of "The Promotion of Mechanical Properties by Bone Ingrowth in Additive-Manufactured Titanium Scaffolds"

_jfb, 2022, doi:10.3390/jfb13030127_

Round 1

Reviewer 1 Report

Comments and Suggestions for Authors can be found in attachment.

Author Response

Response to the Reviewer’s Comments:

On behalf of all the authors, I would like to thank you for your constructive comments on our manuscript, which have helped to improve the quality of this manuscript. And the English have been revised throughout the text. The point-by-point responses to your comments are as follow:

Q1:

  1. Thank you for your very professional suggestion. We add a new table in Appendix A, Scheme 2, which describes the number n belonging to each group for each test. (Line 145~Line 146, Line 435~Line 436, words in blue)

Q2:

  1. We apologize for the inaccurate statement. Yes, we did develop a CT-based reconstructed model for the BCC scaffold. The CT-based model aims to compare the predicted mechanical properties of the fitted model to that of the CT-based model as we mentioned in Line 191~line 196. Therefore, the elastic modulus and compressive strength of the CT-based model were demonstrated in Figure 6.

Compared with the “simplified” model, the advantage of the CT-based model is that it accurately reconstructs the micromorphology and structure of the scaffold, while the disadvantage is that obtaining a CT-based model is time-consuming and computationally expensive. (Line 361~Line 363, words in blue)

  1. The aim of the finite element analysis in this manuscript was to investigate the effect of the bone ingrowth in the scaffold on the mechanical performance of the scaffold, the BCC scaffold was used as an example. We are concerned that the comparison of the two kinds of scaffolds would bring up further issues beyond the aim of the finite element analysis if the diamond scaffold was also investigated by the finite element method.

  1. The sentence “based on the homogenization theory of porous structure” was deleted since we realized that simulating the mechanical properties by finite element model is a general methodology, and does not need to be based on homogenization theory. A reference was also added from the previous study of our group. (Line 197~Line 198, Line 202~Line 203, Ref 24, words in blue)

  1. It is precisely because of the unpredictability and morphological complexity of bone ingrowth in the scaffold that the authors propose this phenomenological model to study the potential enhancement of the bone ingrowth on the mechanical properties of the scaffold. Usually, bone tissue is recognized as tending to adhere on the surface of the micro-structs of the scaffold, and that was why the parameter of volume of the wrap region (V) was defined. On the other hand, we assumed that the bone tissue would be able to fulfill the pores of the scaffold long enough after implantation, and we defined the parameter width of the fulfilled region (t) to characterize this situation. However, in the animal experiment in this manuscript, due to the short implantation time, none of the pores in the scaffolds was observed to be fulfilled by bone tissue in either Micro-CT images or histological staining. Thus, a FE model bone of scaffolds whose micro-structs were uniformly wrapped was developed to simulate the experimental results, aiming to validate the bone-scaffold FE model.

  1. “L4(23)”is the representation of an orthogonal test, where 3 denotes the number of factors, which were the type of bone tissue, t and V in this study; 2 denotes the number of levels of each factor, and 4 denotes the number of experimental runs. A description of this orthogonal experimental design was added to the manuscript (Line 255~Line 257, words in blue), and for clarification, a table (Scheme 5) was added in Appendix A to present what parameters were studied as variables in the finite element model (Line 442~Line 444, words in blue).

Q3:

  1. Thanks for the careful review. The threshold of the gray value of the bone tissue in the CT images was 300~1300 and the threshold of the scaffold was set to 1300~4000. The image reconstruction method was added to the manuscript (Line 158~Line 152, words in blue). The missed ordinate of Figure 4E was added, and thanks to the reviewer for the reminder (Line 276~ Line 279, Figure 4E, words in blue).

  1. Yes. The second “Diamond, 2 weeks” was corrected to “Diamond, 4 weeks” in Figure 5A (Line 289).

  1. It has to clarify that the variables t and V were not involved in the study of mechanical properties of the blank scaffolds (i.e. scaffolds with no bone tissue growing into them). All the results in Figure 6 were from the blank scaffolds. After a correction factor C has been obtained, it was always taken into account when studying the effect of bone ingrowth on the mechanical properties of the scaffolds.

  1. Yes, since only the elastic modulus of the Ti alloy was modified by the correction factor C, it does not have any impact on the compressive strength.

  1. The type of bone tissue was indeed investigated in the study, but due to our oversight, the legend was missed in Figure 7G and 7H. Dark pink is cortical bone and light pink is cancellous bone. The missed legend was corrected (Line 324).

  1. Thanks for the reviewer’s question. Developing model of bone tissue by micro-CT or histological images does allow very accurate model to be obtained. However, on the one hand, this option is time-consuming and computationally expensive. On the other hand, the morphology of the bone tissue in any scaffold is case-specific thus making it difficult to quantify the pattern of bone tissue in the scaffolds and further its effect on the mechanical properties of the scaffolds.

Q4:

  1. Some references were provided according to the reviewer’s kind suggestion (Line 353).

  1. We apologize for missing of the legend in Figure 7G and 7H, it was the omission of the legend that led to the reviewer’s misunderstanding. No, this observation was obtained from the finite element results of this manuscript, basically from Figure 7G and 7H.

  1. The present manuscript does provide an inadequate discussion of the gradient porous structure, so any description of the gradient has been removed in order to focus on the central idea.

  1. We believe that it is still the absence of the crucial legend in Figure 7 that prevents the effect of bone tissue type on the mechanical properties of the scaffold from being presented, ultimately leading to the conclusion that the original results do not support it. After Figure 7 has been modified, the conclusions should be reasonably supported by the results.

Q5:

The authors acknowledge the reviewer for the careful review. The issues mentioned by the reviewer in Q5 have been revised or responded. The details are presented as follows. Besides, other typographical errors were also modified.

  1. Modified.
  2. The words “gradient porous implant” were corrected to “porous implant”
  3. Modified.
  4. Modified.
  5. Modified.
  6. Modified.
  7. Here are just few examples for the RVEs which were usually employed in literatures, without intending of summarize all the RVEs in a categorized manner.
  8. Modified.
  9. Modified.
  10. Modified.
  11. Modified.
  12. Modified.
  13. Modified.
  14. Modified.
  15. Modified.
  16. Modified.
  17. The words “Surface area” refers to the sum of the surface areas of all the micro-structs in a scaffold.
  18. Modified.
  19. Modified.

20.It means that the theoretical model which came from the Computation-Aided Design software, but not reconstructed by any reverse engineering technology.

  1. Modified.
  2. Modified.
  3. Apologies for the misleading expressions. The sentences were modified to clarify the methods used in the calculation of elastic modulus and strength (Line 207~Line 209).
  4. Modified.
  5. Modified.
  6. Modified.
  7. Modified.
  8. These sentences were deleted.
  9. The material Ti6Al4V is the titanium alloy used in the manuscript, for consistency, it was changed to “Ti-alloy”. The material properties of PEEK were deleted. The references of the material properties of bone was added (Ref 29, 30).
  10. Modified.

Reviewer 2 Report

The subject of the manuscript “The Promotion of Mechanical Properties by Bone Ingrowth in the Additive Manufactured Titanium Scaffold” by Sun C. et al. is focused on the investigation of the effect of the bone ingrowth on the mechanical properties of the osseointegrated implants through animal experiments and computational simulation.

One major drawback of this manuscript is the used English. There are a lot of grammatical errors in the whole text. A small part of them was highlighted by the Reviewer (see attached pdf), but the authors should pay attention to this aspect when submitting the revised version of the manuscript. In this respect, either a native English speaker or a professional English editor must be addressed.    

The manuscript can be accepted for publication after the authors will address all the raised queries (in the order they appear in the manuscript):

1. “its influence” (page 2, lines 48 to 49) – who’s influence?

2. “The limited …….interface” (page 2, lines 53 to 55) – the authors should rephrase this.

3. “pore sizes of” (page 2, line 56) – what is the dimension??

4. “finite element (FE)” (page 3, line 109) – once an acronym was defined (page 2, line 77), it should be further used.

5. The authors should clearly explain in the “Introduction” section what exactly their study adds to the literature that was not already there. They should elaborate also on what makes their study unique, original, and novel.

6. What were the group sizes used in this study? This info is important and should appear in the main text.

7. Did the authors take into consideration the rule of the “3Rs” when deciding on the total number of rabbits? The authors should explain.

8. What was the value of the correction factor C? (page 6, line 199).

9. “For deeper……uniformly wrapped.” (page 8, lines 224 to 231) – this phrase is extremely long and therefore hard to follow and understand. The authors are suggested to split it into at least 3 small phrases.

10. Statistical analysis plays a very important role in this type of experiments and therefore a special section dedicated to statistics should be introduced in the manuscript.

11. “images of the scaffolds with different scaffolds” (page 9, line 249) – should be rephrased.

12. The magnification bar in Figure 4A-D is not visible.

13. In Figures 4-6, the authors should explain what are the used error bars: standard error, standard deviation, etc. This explanation should be introduced in the legend of the figure.

14. In the inset of Figure 5A, “Diamond 2 Weeks” should read “Diamond 4 Weeks”.

15. “by assuming that ….. the scaffolds” (page 12, line 294) – should be rephrased.

16. “However ….. ingrowth” (page 15, lines 370 to 373) – this phrase is too long and it should be splitted into at least 2 small phrases.

17. “4. Discussion” (page 16, lines 387 to 391) – this Reviewer does not understand the meaning of this section. It should be therefore deleted from the main text.

Author Response

Response to the Reviewer’s Comments:

On behalf of all the authors, I would like to thank you for your constructive comments on our manuscript, which have helped to improve the quality of this manuscript. The issues highlighted by the reviewers in the PDF have been corrected one by one and the English have been revised throughout the text. The point-by-point responses to your comments are as follow:

  1. “its influence” was corrected to “their influence”. It means the influence of pore size and porosity (Line 48, words in red).

  1. This sentence was rephrased to “The available researches characterizing the tensile bonding strength and the shear bonding strength of the bone-implant interface are limited.” according to the reviewer’s suggestion (Line 53~Line 55, words in red).

  1. The sentence was rephrased to “The tensile bonding strength between Titanium-alloy (Ti-alloy) scaffolds with RVEs of diamond architecture with different pore sizes was measured by Taniguchi et al”. It is therefore no longer necessary to indicate the dimension of the pore sizes (Line 55~Line 57, words in red).

  1. Thanks for the reminder from the reviewer. All the “finite element”, after the acronym was defined, were corrected to “FE” (Line 113, 233, 361, words in red).

  1. Based on reviewer’s suggestion, the author explain the innovative of this study in both section of Introduction and Discussion (Line 102~ Line 108, Line 340~Line 345, words in red).

  1. The detail of the group size was presented in Appendix A, scheme 2 (Line 415~416, words in blue).

  1. Yes, the principles of 3Rs were followed in the design of the animal experiments. The information on the ethics approval and 3Rs was stated in the manuscript (Line 133~Line 136, word in red).

  1. The correction factor C was 0.39, which was indicated in Line 304.

  1. Yes, we agree with the reviewer’s comments. The sentence was rephrased (Line 244~Line 253, words in red).

  1. A new section 2.3.5 was added based on the reviewer’s very kind suggestion (Line 181~Line 185, words in red).

  1. The second “scaffolds” was corrected to “RVEs” (Line 272, words in red).

  1. A legend bar was added in the Figure 4A. For figure 4A~4D, the same magnification was used (Line 276).

  1. All the error bars represented standard deviation. In accordance with your comments, this has been declared in the new section of 2.3.5 on statistics.

  1. Sorry for the mistake. It was corrected (Line 289).

  1. The sentence “by assuming that… the scaffolds” was deleted since the motivation for developing this model has already been set out in the section of 2.3.3.

  1. The long sentence was rephrased into two sentences according to the reviewer’s suggestion. (Line 403~Line 406, words in red)

  1. Yes it should be deleted.

Round 2

Reviewer 1 Report

Comments and Suggestions for Authors can be found in attachment.

Reviewer 2 Report

The authors took into consideration all the queries raised by the Reviewer.

Unfortunately, despite the fact that the authors state that they used a professional editor for the used English, typos and other grammatical errors are still present in the manuscript. The authors should pay attention to this important aspect before resubmission.

Author Response

On behalf of all the authors, I would like to thank you for your kind comments on our manuscript. We realized that there are some typos and grammatical errors still remains in the manuscript even it was been edit by the professional English editor. With the help of the our coauthors from University College London, the manuscript was revised to eliminate the spelling and grammatical errors as much as possible. All the correction were marked in green words in the revised manuscript. We hope that this revision will meet the standards of the Journal of Functional Biomaterials.

Round 3

Reviewer 1 Report

Comments and Suggestions for Authors can be found in attachment 
